# Sleep Fragmentation Accelerates Carcinogenesis in a Chemical-Induced Colon Cancer Model

**DOI:** 10.3390/ijms24054547

**Published:** 2023-02-25

**Authors:** Da-Been Lee, Seo-Yeon An, Sang-Shin Pyo, Jinkwan Kim, Suhng-Wook Kim, Dae-Wui Yoon

**Affiliations:** 1Department of Health and Safety Convergence Science, Graduate School, Korea University, Seoul 02841, Republic of Korea; 2Sleep Medicine Institute, Jungwon University, Goesan-gun 20204, Chungcheongbuk-do, Republic of Korea; 3Next&Bio Inc., Techno Complex Building 6F, Korea University, Seoul 02841, Republic of Korea; 4Department of Biomedical Laboratory Science, Jungwon University, Goesan-gun 20204, Chungcheongbuk-do, Republic of Korea; 5BK21 FOUR R&E Center for Learning Health Systems, Korea University, Seoul 02841, Republic of Korea

**Keywords:** carcinogenesis, colon cancer, reactive oxygen species, sleep fragmentation

## Abstract

Aims of this study were to test whether sleep fragmentation (SF) increased carcinogenesis and to investigate the possible mechanisms of carcinogenesis in a chemical-induced colon cancer model. In this study, eight-week-old C57BL/6 mice were divided into Home cage (HC) and SF groups. After the azoxymethane (AOM) injection, the mice in the SF group were subjected to SF for 77 days. SF was accomplished in a sleep fragmentation chamber. In the second protocol, mice were divided into 2% dextran sodium sulfate (DSS)-treated, HC, and SF groups and were exposed to the HC or SF procedures. Immunohistochemical and immunofluorescent stainings were conducted to determine the level of 8-OHdG and reactive oxygen species (ROS), respectively. Quantitative real-time polymerase chain reaction was used to assess the relative expression of inflammatory and ROS-generating genes. The number of tumors and average tumor size were significantly higher in the SF group than in the HC group. The intensity (%) of the 8-OHdG stained area was significantly higher in the SF group than in the HC group. The fluorescence intensity of ROS was significantly higher in the SF group than the HC group. SF accelerated cancer development in a murine AOM/DSS-induced model of colon cancer, and the increased carcinogenesis was associated with ROS- and oxidative stress-induced DNA damage.

## 1. Introduction

In recent years, evidence about the significant association between sleep disorders and cancer has accumulated. Many studies have demonstrated that various types of sleep disorders, such as circadian rhythm sleep-wake disorders, insomnia, hypersomnia, and sleep-related breathing disorders, are risk factors for cancer. However, systematic reviews and meta-analyses of the association between sleep duration and cancer risk have shown that the increase in cancer risk due to altered sleep duration is limited to specific ethnicities and differs by cancer type [1], suggesting that different sleep problems might have different effects on cancer risk.

Sleep fragmentation (SF) is frequently found in patients with obstructive sleep apnea (OSA), periodic limb movements, insomnia, and chronic pain. It can also be induced by environmental factors such as light, noise, and inappropriate temperature or humidity during sleep. The influence of SF, which can be defined as frequent disruption of sleep architecture despite the optimal duration of sleep [2], on cancer incidence or adverse prognosis has been investigated relatively little, possibly due to limitations on human study. However, experimental studies using mice showed that fragmented sleep accelerates tumor growth and progression through the recruitment of tumor-associated macrophages and the reduction of nicotinamide adenine dinucleotide phosphate (NADPH) oxidase type 2 activity [3,4]. Those studies investigated the effects of SF on tumors inoculated in mice, but the effects of SF on carcinogenesis remain unexamined. For this study, we hypothesized that chronic SF can accelerate tumor development as well as tumor growth. To test our hypothesis, we used chemical-induced colon carcinogenesis models with azoxymethane (AOM) and dextran sodium sulfate (DSS). We chose this colon cancer model using a carcinogen and a colitis-inducing substance for two main reasons. First, previous human studies reported that OSA, a common sleep-breathing disorder that causes SF, is significantly associated with colorectal neoplasia [5]. Second, it takes too long to observe spontaneous carcinogenesis in an animal model of sleep disturbance without genetic modification or the administration of a carcinogen. Even intermittent hypoxia (IH), a more severe model than SF that also mimics OSA, took more than 3 months to induce spontaneous carcinogenesis in old mice (15 months).

In this study, we used two different experimental protocols. In the first protocol, we explored the effect of SF on chemical-induced colon carcinogenesis and its possible mechanisms using macroscopic examinations of colon tumors and histological and molecular analyses. In the second protocol, we conducted an additional experiment to test the possible mechanisms hypothesized from the results of the first protocol.

## 2. Results

### 2.1. Comparison of Tumor Number and Size between the HC and SF Groups

In Experiment 1, we compared the tumor number and size between the HC and SF groups after sacrificing the mice. The total number of tumors was significantly higher in the SF group than in the HC group (Figure 1A,B). We classified tumor size as <2 mm or >2 mm. The number of tumors larger and smaller than 2 mm was both significantly higher in the SF group than in the HC group (Figure 1C,D). The tumors were also larger in the SF group than in the HC group (Figure 1E). Inflammation is known to shorten colon length, so we measured colon length to assess whether SF exacerbated the inflammatory status of the colon in the AOM/DSS-treated mice. The colon length did not differ significantly between the HC and SF groups (8.9 ± 1.7 vs. 9.2 ± 2.0) (Figure 1F,G).

### 2.2. Histological Assessments of Colon

The AOM/DSS colon carcinogenesis model is based on the induction of colitis. Therefore, we first hypothesized that the increased carcinogenesis caused by SF was due to the exacerbation of colitis. In the H & E stain results, however, we found no significant difference in the inflammation score between the HC and SF groups (1.3 ± 0.2 vs. 1.5 ± 0.3) (Figure 2A,B). Next, we performed IHC staining to assess the level of 8-OHdG, a well-known oxidative stress-induced DNA damage marker, because a previous study reported that sleep deprivation specifically increased ROS in the colon [6]. ROS is also an important cause of carcinogenesis [7,8]. Spearman’s correlation analysis also showed a significant positive correlation between 8-OHdG intensity and the number of tumors in Figure 2C (r = 0.731; *p*-value = 0.0004). The intensity of the 8-OHdG stained area (%) was higher in the SF group than in the HC group (Figure 2D,E). These IHC and correlation analysis results imply that the increased tumor development caused by SF was associated with elevated 8-OHdG levels in the colon.

### 2.3. ROS Measurements in Colon Tissue

To evaluate whether SF itself can elevate oxidative stress in the colon, we designed an SF experiment without AOM/DSS treatment (Experiment 2). The confocal microscope analysis showed that the fluorescence intensities of ROS in the colon were significantly higher in the SF group than in the HC group (Figure 3A,B).

### 2.4. Gene Expression Analysis

Next, we conducted qRT-PCR to assess the gene expression associated with ROS generation and inflammation in the colon (Experiment 2). The level of *NOX1*, a colonic epithelium-specific ROS-generating enzyme, tended to increase in the SF group, but the difference from the HC group did not reach statistical significance (Figure 4), possibly due to our small sample size. The level of *NOX2*, a macrophage-specific ROS-generating enzyme, did not differ between the groups. No significant differences between the HC and SF groups were found in proinflammatory cytokine (*IL-1β* and *IL-6*) levels, supporting the results of Experiment 1 that SF increased oxidative stress in the colon without exacerbating inflammation.

## 3. Discussion

This chemical-induced carcinogenesis study has shown that chronic SF increases colon carcinogenesis in mice. Tumor size was also larger in the colons of SF mice than in the control mice. Elevated ROS and the resulting oxidative DNA damage were associated with SF-induced carcinogenesis. To our knowledge, no previous studies have examined the direct effect of SF on carcinogenesis; therefore, our results support the previous finding that OSA is associated with an increased risk of colon cancer and further suggest that SF alone might contribute to cancer development even in the absence of confounding conditions such as sleep duration or IH.

Epidemiology studies about the association between sleep disorders and cancer have already shown that chronically altered sleep cycles [9] or short sleep duration [10] increase colorectal cancer risk. Sleep quality is also an important factor in cancer risk. The English Longitudinal Study of Ageing examined the association between sleep quality and incident cancer risk and found that poor sleep quality, as assessed by questionnaire, was associated with an increase in the long-term incident cancer risk (HR 1.586, 95% CI 1.149–2.189) [11]. Increased production of glucocorticoids, activation of the sympathetic nervous system, and exacerbated inflammation have been suggested as possible mechanisms linking sleep disorders and cancer pathogenesis [12,13], but the causal relationship between them is still ambiguous due to a lack of appropriate experimental models and design limitations in human studies.

SF is a common feature of many types of sleep disorders, and it is particularly characteristic of patients with OSA, in whom the cessation of breathing causes arousal during sleep, which opens the airway but interferes with sleep continuation. These arousals can occur several to dozens of times per hour during sleep.

Most human and animal studies that have investigated the influence of SF on body systems have focused on cognitive function, the cardiovascular system, or metabolism. They have reported a significant causal effect of SF or cross-sectional relationships between SF and poor cognitive function [14], endothelial dysfunction [15], atherosclerosis [16], and insulin resistance [17]. To the best of our knowledge, only two previous animal studies examined the effect of SF on tumors. Hakim et al. used mice deficient in TLR4 or its downstream molecules MYD88 or TRIF [3]. After the engraftment of tumor cells, the mice were subjected to SF for 28 days and examined for tumor growth and invasiveness. SF accelerated tumor growth and increased tumor invasiveness to adjacent tissue through changes in the microenvironment that facilitated cancer progression. The other study that examined the effect of SF on tumors also reported increased tumor growth in mice subjected to SF and that increased tumor growth was mediated by phagocytic Nox2 activity within the tumor [4]. In those two studies, however, tumor cells were implanted into mice before SF. Therefore, those studies cannot confirm the carcinogenic effect of SF.

We found elevated ROS in the colon tissue of mice subjected only to SF. Although it is difficult to say that the increased ROS found in the AOM/DSS experiments was the direct or sole cause of SF-induced colon carcinogenesis because we did not perform an intensive mechanistic study using pharmacological inhibition or genetic manipulation experiments, it is well known that ROS-induced genomic instability, such as 8-OHdG accumulation, is an important cause of carcinogenesis [18,19].

Many studies have reported altered or increased oxidative stress in different organs during sleep disturbance. In rodents, antioxidant defense responses to sleep loss were decreased in the liver [20], and ROS were elevated in hepatocytes by REM sleep deprivation [21], implying that the liver is susceptible to sleep loss. Sleep loss also alters antioxidant responses in the brain [22,23]. However, it is not known whether the increased oxidative stress caused by sleep disturbances is a cause or a result of other damage.

ROS accumulation in the gut is a critical factor in death caused by sleep deprivation (SD). Vaccaro et al. showed that SD induced ROS accumulation, oxidative stress, and DNA damage only in the guts of flies and mice [6]. Elevated oxidative stress shortened the lifespans of both animals, and gut-specific expression of antioxidant enzymes through genetic manipulation extended the survival of both. These findings indicate that the detrimental effects of SD are not equally distributed to all organs and are concentrated in the gut in terms of oxidative stress. Although the sleep disturbance model used in Vaccaro’s study differed slightly from our model (SF vs. SD), our results support Vaccaro’s findings and further suggest a possible role of SF in colon carcinogenesis.

## 4. Materials and Methods

### 4.1. Experimental Design

Thirty-seven male, 8-week-old C57BL/6 mice (DBL Co., Ltd., Eumseong, Republic of Korea) were used in this study, which consisted of two different protocols (Figure 5A,B). The first protocol (Experiment 1) examined the effect of SF on colon carcinogenesis. The mice in Experiment 1 were randomly divided into two groups: home cage control (HC; n = 12) and SF (n = 12). Mice in both groups received one intraperitoneal injection of AOM (10 mg/kg; Sigma-Aldrich, St. Louis, MO, USA). Then, the mice in the HC and SF groups were exposed to normal conditions and the SF protocol, respectively, for 77 days. 2% DSS (MP Biomedicals, Santa Ana, CA, USA) was given in the drinking water on Days 7–14, Days 28–35, and Days 49–56. The mice were sacrificed on Day 77, and tissue samples were harvested for further analysis. The second protocol (Experiment 2) tested our hypothesis that SF alone can elevate reactive oxygen species (ROS). Mice in the HC (n = 5) and SF groups (n = 5) were maintained in normal conditions and the SF chamber for 4 weeks without any administration of AOM or DSS. The 2% DSS group (n = 3) was given 2% DSS in their drinking water between 3 and 4 weeks as a positive control of the ROS measurements. Both experimental protocols were approved by the Jungwon University Institutional Animal Use and Care Committee and are in close agreement with the NIH Guide on the Care and Use of Animals.

### 4.2. Sleep Fragmentation

To induce SF, we used a commercially available SF chamber (Model 80391; Lafayette Instrument, Lafayette, IN, USA). The chamber has a sweeping bar inside that moves from one end to the other. When it touches the mice, they awaken from sleep. To mimic the arousal frequency shown in patients with severe sleep apnea (30 awakenings/h), the sweeping bar was set to move once every 2 min. Although we did not perform simultaneous electroencephalography monitoring to evaluate whether SF was appropriately induced, several studies using the same chamber have already validated this SF model [24,25].

### 4.3. Histological Evaluation

Colon tissue was harvested on Day 77 and washed with cold phosphate-buffered saline (PBS). To compare the colon length between the HC and SF groups, colons from the cecum to the rectum were placed on a plate with a black background and photographed with a digital camera. After taking those colon images, the cecum was removed, and the remaining colon was opened longitudinally along the main axis to expose the internal tumors. After taking a picture of the inside of each colon, the colons were divided into their distal parts and middle parts for histological analysis and quantitative real-time polymerase chain reaction experiments, respectively. Tumor number and size were evaluated by two independent investigators and expressed as the mean value of the two measurements. To obtain reproducible results, the number of tumors was counted by dividing them into those that were 2 mm or larger and those that were 2 mm or smaller [26]. Hematoxylin and eosin (H & E) staining was performed for general histological assessments and colonic inflammation score calculation. After H & E staining, five random areas of the colon per tissue slide were used to calculate the colonic inflammation score. Colonic inflammation was classified into five grades (Grade 0 to Grade 4) based on the criteria of Cooper et al. [27,28].

### 4.4. Immunofluorescent ROS Staining

Immunofluorescent staining of colon tissue from mice in the HC and SF groups in Experiment 2 was conducted to measure ROS using a protocol from a previous study [6]. Distal colons excised from the mice were immediately placed on dry ice and embedded in the O.C.T. compound (Tissue-Tek). 30-μm thick sections were obtained using a cryomicrotome. The sections were air-dried and then incubated with 10 μM dihydroethidium (Sigma-Aldrich, St. Louis, MO, USA) at 37 °C for 30 min. After being washed with PBS, the sections were incubated with 1 μg/mL Hoechst 33342 (Thermo Fisher, Waltham, MA, USA) at room temperature for 10 min. Then, the sections were mounted with Fluoroshield (Sigma-Aldrich, St. Louis, MO, USA). The red fluorescence of each image was obtained by excitation at 610 nm and collected at 600–780 nm using confocal microscopy (LSM 700, Carl Zeiss, Jena, Germany).

### 4.5. Immunohistochemical 8-OHdG Staining

For immunohistochemistry (IHC), excised colon tissue was fixed with 10% neutral-buffered formalin, embedded in paraffin, and cut into 4-μm slices using a rotary microtome. For antigen retrieval, tissue slides were autoclaved with HIER citrate buffer (pH 6.0; Zytomed Systems, Berlin, Germany). IHC for 8-OHdG staining was conducted using a commercial mouse and rabbit-specific HRP/DAB detection kit (Abcam, Cambridge, UK). After being incubated with 3% hydrogen peroxide in methanol for 10 min, the sections were treated with a blocking solution for 1 h at room temperature to remove nonspecific background reactions. Following treatment with blocking solution, the sections were incubated with anti-8-OHdG/8-oxo-dG monoclonal antibody (1:200; JalCA, Nikken SEIL CO, Shizuoka, Japan) for 1 h at 4 °C. After washing the samples with TBST four times, a biotinylated goat anti-polyvalent was applied to the tissues and incubated for 10 min at 4 °C. After being treated with streptavidin peroxidase for 10 min, the sections were developed with 3,3′-diaminobenzidine solution. As a counterstain, hematoxylin QS (Vector Laboratories, Inc., Burlingame, CA, USA) was applied to the tissues for 1 min, and then the sections were dehydrated, cleared, and mounted for microscopic examination. The stained slides were scanned with a Pannoramic SCAN II slide scanner (3d Histech, Budapest, Hungary). Then, 4–5 high-power field images (400 X) per slide were randomly acquired. The intensity (%) of the stained area in each image was blindly analyzed using ImageJ (NIH, Bethesda, MD, USA).

### 4.6. Quantitative Real-Time Polymerase Chain Reaction (qRT-PCR)

Total RNA was extracted from the colon tissue using an RNeasy mini kit (Qiagen, Germantown, MD, USA). One microgram of total RNA was used for cDNA synthesis and reverse transcribed using a Tetro cDNA synthesis kit (Bioline, London, UK). qRT-PCR was performed using a StepOne Plus^TM^ real-time PCR system (Applied Biosystems, Waltham, MA, USA). TaqMan real-time PCR master mixes were used for TaqMan assay–based real-time PCR. The following TaqMan primer and probes were purchased from Applied Biosystems: NADPH oxidase 2 (NOX2) assay# Mm01287743_m1, NOX1 assay# Mm00549170_m1, interleukin (IL)-1β assay# Mm00434228_m1, IL-6 assay# Mm00446190_m1, and β-actin assay# Mm02619580_g1. All reactions were performed in duplicate. The 2ΔΔ method was used for the relative comparison of genes between the HC and SF groups. β-actin was used as a housekeeping gene for normalization.

### 4.7. Statistical Analysis

Data are expressed as the mean ± standard error of the mean (S.E.M). Differences in the means were evaluated using the non-parametric Kruskal-Wallis test. Mann-Whitney U-tests were used for the post hoc analyses and comparisons of variables between two independent groups. Spearman’s rank correlation coefficients were calculated to evaluate correlations between the intensity of 8-OHdG stained areas (%) and the number of tumors. All statistical analyses were performed using IBM SPSS version 21.0 (SPSS; Chicago, IL, USA), and a *p*-value < 0.05 was considered to be statistically significant.

## 5. Conclusions

We found that SF has carcinogenic effects in the murine AOM/DSS colon cancer model. Moreover, elevated oxidative stress-induced DNA damage correlated with an increase in carcinogenesis. Our findings suggest that SF is a risk factor for colon cancer and that preventing ROS accumulation is important for preventing the development of SF-induced colon cancer. Further studies are needed to elucidate whether SF can induce carcinogenesis in other visceral organs and determine the exact mechanism by which SF promotes carcinogenesis.

## Figures and Tables

**Figure 1 ijms-24-04547-f001:**
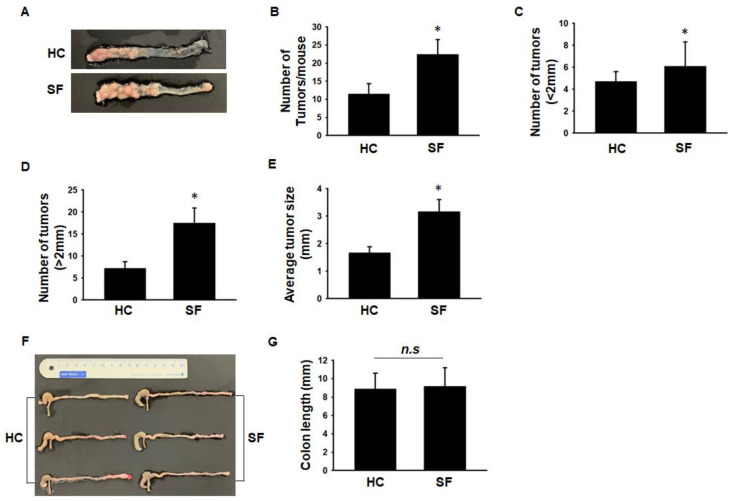
Tumor growth in mice in the home cage control (HC) and sleep fragmentation (SF) groups. (**A**). Macroscopic view of colon tumors that developed in the mice of the HC and SF groups. (**B**). Comparison of the number of tumors. (**C**). Comparison of the number of tumors < 2 mm. (**D**). Comparison of the number of tumors > 2 mm. (**E**). Average tumor size in the HC and SF groups. (**F**). Representative images of mouse colons from the HC and SF groups. (**G**) Comparison of colon lengths between the HC and SF groups. Data are presented as the mean ± S.E.M. * *p* < 0.05 compared with the HC group, as determined by the Mann-Whitney U test. n.s. non-significant.

**Figure 2 ijms-24-04547-f002:**
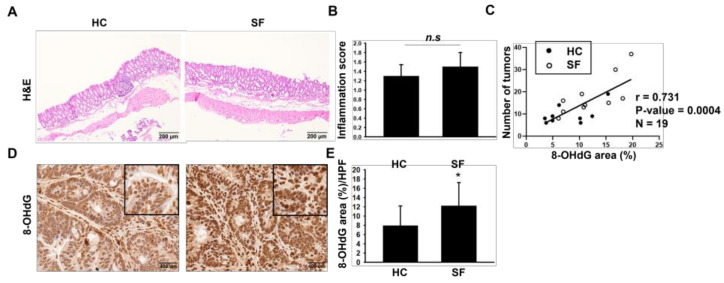
Results of H & E staining and immunohistochemical staining for 8-OHdG. (**A**) Representative H & E-stained images of colon tissue and (**B**) comparison of inflammation scores. (**C**) Spearman’s correlation analysis between number of tumors and 8-OHdG area (%). (**D**) Representative immunohistochemical-stained images for 8-hydroxyl-2′-deoxyguanosine (8-OHdG) in the colons of mice from the HC and SF groups and (**E**) quantification of the stained area (%). Data are presented as the mean ± S.E.M. * *p* < 0.05 compared with the HC group, as determined by the Mann-Whitney U test. n.s. non-significant.

**Figure 3 ijms-24-04547-f003:**
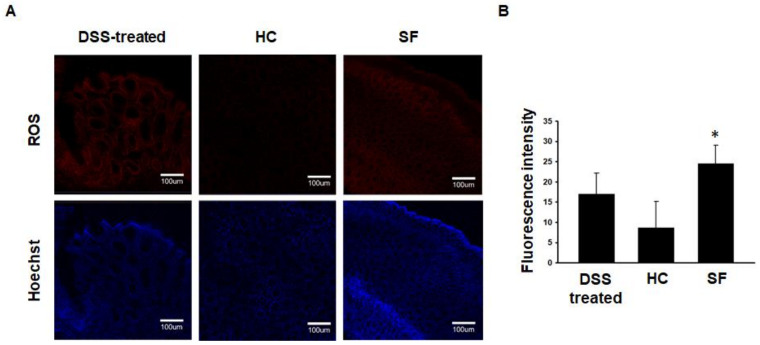
Confocal microscopic images showing ROS in colon tissue from mice in the DSS-treated, HC, and SF groups. (**A**). Representative confocal images of ROS in the colons of mice in the 2% DSS-treated (n = 3), HC (n = 5), and SF (n = 5) groups. (**B**). Comparison of fluorescence intensity showing ROS in colon tissues across the groups. Data are presented as the mean ± S.E.M. * *p* < 0.05 compared with the HC group, as determined by the Mann-Whitney U test.

**Figure 4 ijms-24-04547-f004:**
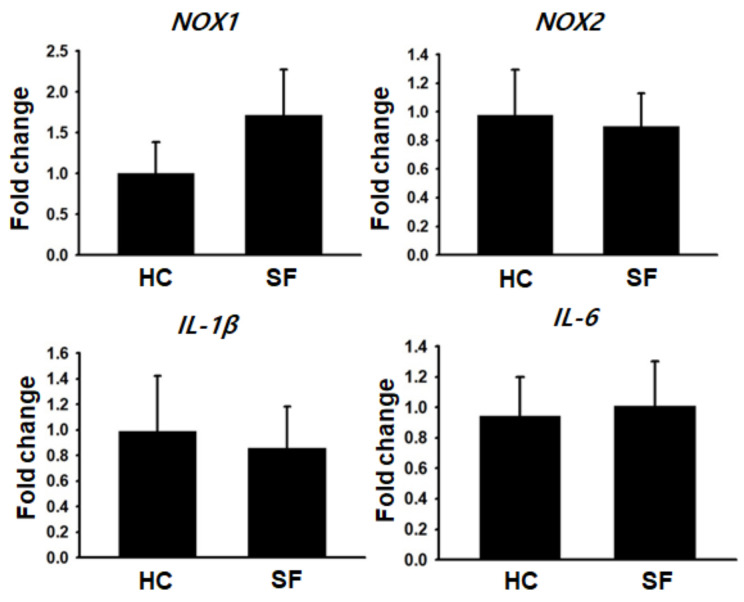
Expression of genes related to ROS generation and inflammation in tumor tissues from mice in the HC and SF groups. Comparison of relative NOX1, NOX2, IL-1β, and IL-6 mRNA expression levels in tumors from mice in the HC and SF groups. Data are presented as the mean ± S.E.M3.

**Figure 5 ijms-24-04547-f005:**
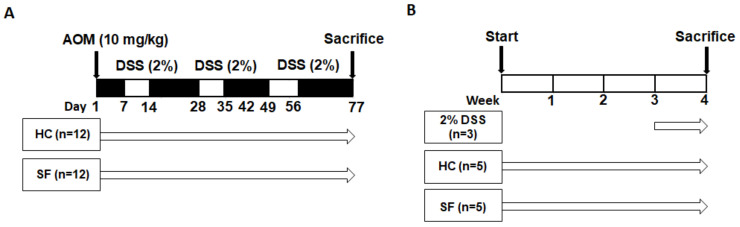
Schematic representation of the experimental protocols. (**A**). Schematic design of Experiment 1. A single intraperitoneal injection of 10 mg/kg azoxymethane (AOM) was given to mice in both the control and sleep fragmentation (SF) groups. Subsequently, mice in the control group were maintained in a normal cage until the end of the experiment (77 days). Mice in the SF group were subjected to SF, which was accomplished by a sweeping bar activated every 2 min (30/h). Dextran sodium sulfate (DSS) was given in the drinking water on Days 7–14, 28–35, and 49–56. (**B**). Schematic design of Experiment 2. Mice in the control and SF groups were maintained in normal cages and SF chambers, respectively, for 4 weeks. 2% DSS was given to other mice in their drinking water at 3–4 weeks as a positive control.

## Data Availability

The data presented in this study are available on request from the corresponding author.

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
