# Peer review of "Sleep Fragmentation Accelerates Carcinogenesis in a Chemical-Induced Colon Cancer Model"

_ijms, 2023, doi:10.3390/ijms24054547_

Round 1

Reviewer 1 Report

-keywords should appear in alphabetical order.

-I do not understand the number of mice because it says that 36 have been used, but there is a group of 12, another of 12, 5 +5+3, that is, 37 instead of 36.

Author Response

Question 1: Keywords should appear in alphabetical order.

Answer: We thank the reviewer for the kind comments. Keywords have been rearranged alphabetically. Please see line 31 of the revised manuscript.

Question 2: I do not understand the number of mice because it says that 36 have been used, but there is a group of 12, another of 12, 5 +5+3, that is, 37 instead of 36.

Answer: We apologize for the confusion and thank the reviewer for the valuable comments. We changed the total number of animals from thirty-six to thirty-seven in 4.1 Experimental design section. Please see line 224 and Figure 5 of the revised manuscript.

Reviewer 2 Report

This study tested whether SF affects colon carcinogenesis in mice induced by AOM/DSS. The first experiment showed that SF increased the number and size of tumors compared to the control group, without any evidence of increased inflammation, but with increased markers of oxidative stress-induced DNA damage (8-OHdG). The second experiment tested the effects of SF without concomitant administration of carcinogens, and found evidence of increased ROS without changes in inflammatory cytokine levels.

Comments

It would have been interesting to study the type of tumor-associated macrophages. Could these data be added?

My impression is that the discussion regarding the effects of OSA should be lighter. It is true that OSA causes SF, but it also causes IH which is a potent factor inducing oxidative stress. The experiments nicely performed in this study provide no information to be directly translated to the OSA situation. This would require additional experiments, with chemical carcinogenesis+IH, and + both IH and SF. I would strictly stay within the SF topic, and amend any part of the discussion speculating on OSA.

It is interesting that SF causes fluorescence intensity higher than that of DSS. Why was only DSS used, without 8-OHdG? Could the lower response be secondary to the change in the model?   

Author Response

Question 1: It would have been interesting to study the type of tumor-associated macrophages. Could these data be added?

Answer: We really thank the reviewer for the valuable comments. As mentioned by the reviewer, preferential M2-type tumor-associated macrophages (TAMs) emerged in the tumors of sleep-fragmented mice and were associated with accelerated tumor growth and progression (Hakim F et al. Cancer Res 2014). Therefore, we sought to find a possible mechanism different from previous study in which sleep fragmentation accelerates carcinogenesis. At present, we did not isolate TAMs from tumor masses at sacrifice. Thus, TAMs data cannot be added to this project. We hope the reviewer understand this situation.    

Question 2: My impression is that the discussion regarding the effects of OSA should be lighter. It is true that OSA causes SF, but it also causes IH which is a potent factor inducing oxidative stress. The experiments nicely performed in this study provide no information to be directly translated to the OSA situation. This would require additional experiments, with chemical carcinogenesis+IH, and + both IH and SF. I would strictly stay within the SF topic, and amend any part of the discussion speculating on OSA.

Answer: We are deeply grateful to the reviewer for these valuable comments and fully agree with them. As recommended by the reviewer, we have removed some of the discussion speculating on OSA corresponding to reference 23~27. Please see line 190~200 of the revised manuscript.

Question 3: It is interesting that SF causes fluorescence intensity higher than that of DSS. Why was only DSS used, without 8-OHdG? Could the lower response be secondary to the change in the model? 

Answer: We thank the reviewer for the helpful comments. Inflammation is a well-known cause of ROS. Inflammation-induced macrophages and neutrophils infiltrating the intestine can produce ROS, which leads to more severe oxidative stress and inflammation (Wang Y et al. J Fun Foods 2020). DSS has been widely used as an inducer of colitis in mice. A previous study reported that administration of 2.5% DSS to 8-week-old C57BL/6 mice for 7 days increased intestinal inflammation, as evidenced by bleeding colons, enlarged spleen, superficial inflammation, epithelial erosion, and immune cell infiltration (Chassaing B et al. Curr Protoc Immunol 2015). Therefore, we thought that administration of 2% DSS in drinking water for 1 week was sufficient to induce colitis and consequently ROS elevation.

In addition, 8-OHdG is not an ROS inducer, but a peroxidation product produced by ROS damaging the guanine base pairs of intracellular DNAs. Therefore, we used only DSS to elevate ROS in mouse colon tissue by stimulating colonic inflammation.

We were also surprised to find higher fluorescence intensity in SF than in DSS. It has been known that severity of colitis depends on DSS concentration, duration, mouse strain, source, body weight. Since the 2% DSS concentration used in our study was lower than the 2.5% DSS of Chassaing B’s study, milder inflammation and consequently weaker ROS production than expected are likely to occur. Therefore, from our experiment, we can say that 4 weeks of exposure to SF induced more ROS generation than 1-week of 2% DSS in mice. We hope the reviewer understand that the purpose of this experiment was not to compare ROS levels between 2% DSS and SF.

Reviewer 3 Report

In this study, Da-Been Lee et al. , used a mouse model of chemical-induced carcinogenesis to explore the effect of sleep fragmentation to colon cancer development. Macroscopic examinations of colon tumors and histological and molecular analyses show that chronic sleep fragmentation may increase colon carcinogenesis in mice. Although many studies have reported altered or increased oxidative stress in different organs during sleep disturbance this study examined the direct effect of SF on colon carcinogenesis. This is a clear, concise, and well-written manuscript. The methods used in the study and the results are well described and discussed. Overall, the results provide a robust conclusion that in an animal model of chemical-induced carcinogenesis of colon cancer sleep fragmentation accelerated cancer development and the increased carcinogenesis may be associated with ROS- and oxidative stress–induced DNA damage.

Author Response

Comments: In this study, Da-Been Lee et al. used a mouse model of chemical-induced carcinogenesis to explore the effect of sleep fragmentation to colon cancer development. Macroscopic examinations of colon tumors and histological and molecular analyses show that chronic sleep fragmentation may increase colon carcinogenesis in mice. Although many studies have reported altered or increased oxidative stress in different organs during sleep disturbance this study examined the direct effect of SF on colon carcinogenesis. This is a clear, concise, and well-written manuscript. The methods used in the study and the results are well described and discussed. Overall, the results provide a robust conclusion that in an animal model of chemical-induced carcinogenesis of colon cancer sleep fragmentation accelerated cancer development and the increased carcinogenesis may be associated with ROS- and oxidative stress–induced DNA damage.

Answer: We are grateful to the reviewer for appreciating our work.  
